# Growth Story of One Diamond: A Window to the Lithospheric Mantle

**Valentin Afanasiev** [1],*, **Sargylana Ugapeva** [2], **Yuri Babich** [1], **Valeri Sonin** [1], **Alla Logvinova** [1], **Alexander Yelisseyev** [1], **Sergey Goryainov** [1], **Alexey Agashev** [1] **and Oksana Ivanova** [1]

1    Sobolev Institute of Geology and Mineralogy, Siberian Branch of the Russian Academy of Science, Koptyuga pr. 3, Novosibirsk 630090, Russia

2    Diamond and Precious Metal Geology Institute, Siberian Branch of the Russian Academy of Science, Lenina Ave. 39, Yakutsk 677000, Russia

*    Correspondence: avp-diamond@mail.ru; Tel.: +7(913)-910-46-95

**Abstract:** A diamond plate cut out of a transparent, colorless octahedral diamond crystal of gem quality, with a small chromite inclusion in the core, sampled from the XXIII CPSU Congress kimberlite (Yakutia, Mirny kimberlite field, vicinities of Mirny city), has been studied by several combined methods: absorption spectroscopy at different wavelengths (UV-visible, near- and mid-IR); photoluminescence, cathodoluminescence, and Raman spectroscopy (local version) and lattice strain mapping; birefringence in cross-polarized light; and etching. The diamond plate demonstrates a complex growth history consisting of four stages: nucleation and growth to an octahedron → habit change to a cuboid → habit change to octahedron-1 → habit change to octahedron-2. The growth history of the diamond records changes in the crystallization conditions at each stage. The revealed heterogeneity of the crystal structure is associated with the distribution and speciation of nitrogen defects. The results of this study have implications for the information value of different techniques as to the diamond structure defects, as well as for the as yet poorly known evolution of the subcontinental lithospheric mantle in the Siberian craton, recorded in the multistage growth of the diamond crystal. At the time of writing, reconstructing the conditions for each stage is difficult. Meanwhile, finding ways for such reconstruction is indispensable for a better understanding of diamond genesis, and details of the lithosphere history.

**Keywords:** diamond; growth history; photoluminescence; cathodoluminescence; Raman spectroscopy; birefringence; etching; crystallization conditions; lithospheric mantle

## 1. Introduction

Diamond, as any other mineral, represents the evolution of the environment in which it was nucleating and growing, and thus bears signatures of the respective changes. Natural diamonds crystallize under high pressures and temperatures at 120 km or deeper in the mantle, and their formation conditions are recorded in diamond-bearing mantle xenoliths transported to the surface with erupting magma. Diamond is one of the few mantle minerals that can preserve heterogeneity features associated with its growth: zones, sectors, fibers, etc. These features show that diamonds commonly grow for quite a long time, in varying physical and chemical conditions [1–6]. Thus, the individual history of a diamond crystal, from its nucleation to eruption with kimberlitic or other diamond-bearing magmas, or to tectonic exhumation (as in the case of the Kumdy Kol complex in Kazakhstan), stores a record of changes in the mantle at respective depth levels. Such a record is commonly poorly pronounced in silicate and ore mantle-derived minerals, which can re-equilibrate under thermodynamic changes during high-temperature mantle processes, but it survives in diamonds and in kimberlitic zircons. The fingerprints of change are preserved in the structure and properties of diamond, despite diffusive contamination with nitrogen and

other impurities or plastic deformation, whereas the growth features of other minerals in kimberlites become erased by re-crystallization. However, the crystallization conditions imprinted in the morphology of diamond crystals only correspond to the final stage of growth.

The heterogeneity of diamond crystals is detectable by different methods, which are more or less sensitive to specific defects, and thus are used according to research objectives. The available methods, informative as to the crystal interior, include old techniques such as etching [1,7,8], anomalous birefringence [9–15], and X-ray topography [16,17], as well as more recent methods of photoluminescence [18,19], cathodoluminescence, trace element analysis of diamond, and ion probe analyses of N and C isotopes, etc. Almost all natural diamonds analyzed by different methods show complex patterns of defects, formed under changeable conditions during their growth. Meanwhile, there are few studies of single crystals that combine several different methods to study the internal inhomogeneity of a crystal.

We are trying to bridge this gap by analyzing a diamond plate by absorption spectroscopy at different wavelengths (UV-visible, near- and mid-IR), photoluminescence, cathodoluminescence, Raman spectroscopy and lattice strain mapping, birefringence in cross-polarized light, and etching. The results allow insights into the stages of diamond growth and furnish new evidence on the information value of different techniques as to the diamond's structural defects. The diamond plate we analyzed demonstrates a complex growth history, with notable changes in crystallization conditions, which have implications for the as yet poorly known evolution of the lithospheric mantle.

## 2. Materials and Methods

### 2.1. Materials

The analyzed sample was cut out of a transparent, colorless octahedral diamond crystal of gem quality, with a small chromite inclusion in the core, and poorly pronounced zoning, from the XXIII CPSU Congress kimberlite (Yakutia, Mirny kimberlite field, vicinities of Mirny city), (Figure 1). The pipe has a diameter of about 50 m, it is a small deposit with a high content of high-quality diamonds. This kimberlite field also hosts well-known diamond deposits—the Mir and Internatsionalnaya pipes.

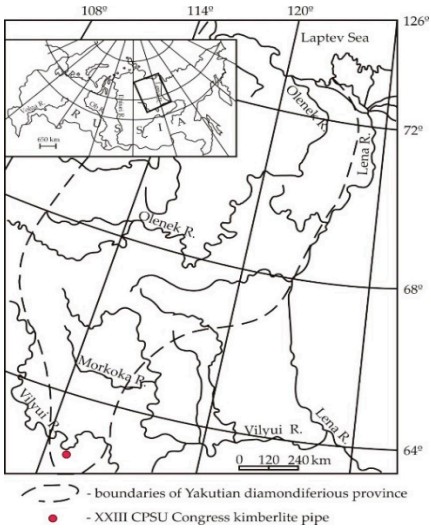

**Figure 1.** Location of the XXIII CPSU Congress kimberlite pipe in Yakutia.

The diamond was cut and polished into a 2.0 mm × 1.8 mm × 0.57 mm plate (sample 4007) parallel to cubic planes (Figure 2).

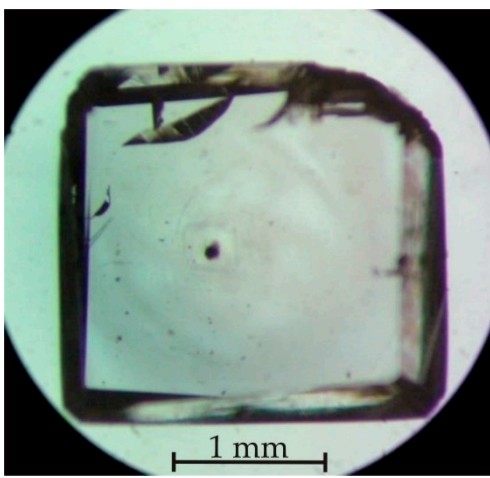

**Figure 2.** Plate 4007 cut out of an octahedral diamond for analysis by different methods.

*2.2. Methods*

Cathodoluminescence. The luminescence was excited at 300–700 nm and imaged using a Centaurus detector and a LEO1430VP scanning electron microscope, operated at an accelerating voltage of 20 keV.

Selective etching. The plate was etched in a $NaNO_3$ melt at 580 °C for 10 min following the procedure reported previously by [20]. The diamond plate was placed in an alundum crucible filled with the melt open to contact with air. The temperature was measured by an S-type Rh/Pt thermocouple (10% Pt), with its junction connected directly to the crucible.

Absorption IR spectroscopy (FTIR). The IR-spectra were obtained on a Bruker Vertex-70 IR-Fourier spectrometer (USA), coupled with a Hyperion 2000 microscope. About 400 spectra of 2 cm$^{-1}$ resolution were recorded with 50 μm square aperture by mapping mode (grid with steps of 122 μm for X,Y-coordinates). The analysis of diamond IR-data with estimation of peak intensities for large amounts of spectra was performed with the help of a specialized batch-processing program IR'nDi-Module [21]. All the spectra in one-phonon region were decomposed into present nitrogen forms, with subsequent calculating of their concentrations using specific coefficients reported in [22,23].

UV-visible and near-IR (200 to 1000 nm) absorption spectroscopy. The spectra were collected at 80° K on a Shimadzu PC 2501 spectrometer at a beam diameter of 300 μm, to a resolution of 0.5 nm.

Photoluminescence spectroscopy. Photoluminescence spectra of 350–800 nm wavelengths were excited by a 350.7 nm pulsed Nd: YLF third harmonic solid-state diode pumped laser and recorded at 80° K with a LOMO SDL1 diffraction luminescence spectrometer at a beam diameter of ~20 μm, to a resolution of 0.5 nm. Samples were placed on a cold finger of the metal cryostat with fused silica windows. The photoluminescence (PL) patterns were obtained under 365 nm UV light from a 100 W mercury arc lamp using a MBS 10 optical microscope (Russia).

Raman spectroscopy. Raman spectra were used to map stress around the chromite inclusion, on a Renishaw in Via (UK) and a Horiba Jobin Yvon LabRam HR spectrometers, using a 532 nm 1–5 mW laser [24]. The local x-y stress field in diamond around the inclusion was evaluated by analyzing the position, broadening, and splitting of the 1332 cm$^{-1}$ ($T_{2g}$) Raman peak of a cubic diamond, at a constant depth z corresponding to the position of the inclusion (z~80 μm distance from the inclusion center to the diamond free surface). The mapping was performed at 1100–1500 cm$^{-1}$ wavelengths, on a 5 μm grid. Signals from the $5 \times 5 \times 5$ μm$^3$ volume were collected at each grid point.

Birefringence was studied on an MBS 10 optical microscope (Russia) equipped with film polaroids.

LA-ICP-MS. The concentrations of elements in diamond were determined by inductively coupled plasma mass spectrometry (LA-ICP-MS) with laser-ablation using a Thermo

Scientific XSERIES2 quadrupole mass spectrometer at the Novosibirsk State University. The mass spectrometer was coupled with a NewWave Research, Nd:YAG 213 nm laser-ablation system. Ablation was performed in a He flux through the sample chamber, using ultra-pure Ar as plasma-forming gas. Cellulose doped with a multi-element solution reported in [25] was used as external standards and $^{13}$C was used as the internal standard. Laser was operated at 10 Hz with a pulse energy of 17–22 mJ·cm$^{-2}$ and a beam size of 100 µm.

Out of a large scope of analyzed elements (Li to U), only a few were selected for further consideration: B, Mg, Al, Ti, Cr, Ni, Cu, Zn, Sr, Y, and Zr, which gave signal at least twice the background intensity; the signals of Sr, Y, and Zr, were very low but detectable against zero background. Among REEs, only La produced a systematic signal corresponding to 0.2–0.5 ppb. All other REE signals were unsuitable, even with zero background; they appeared only sporadically in few analyzed spots. The signals of Si, P, Ca, S, and V, were very strong but the background/signal ratio was only 1.2–1.4, and the data were unreliable because of irremovable background influence. Detection limits strongly depends on background and were 0.01–0.03 ppm for the masses lighter than Sr. For the Sr, Y, and Zr, all the signal collected was considered as meaningful as the measured background was regularly a zero. Doped cellulose standard measured as unknown was reproduced within 5%–20% of recommended values. The low concentrations (~0.8 ppm) glass standard NIST 614 (National Institute of Standards and Technology) were analyzed two times and gave 10% of deviation from recommended values [26] for the elements heavier than Sr, and within 25% for lighter elements. Therefore, the error was accepted as 25% of relative deviation. However, determination of trace elements in diamonds is not a common procedure, and there are no standards that match diamond matrix and composition; therefore, the error at the concentrations lower than 0.1 ppm could be higher.

## 3. Results

### 3.1. Cathodoluminescence

Cathodoluminescence shows the details of the internal inhomogeneity of the diamond well (Figure 3). Cathodoluminescence (CL) patterns record a complex history of crystal growth consisting of four main stages: nucleation and growth to an octahedron → habit change to a cuboid → habit change to octahedron-1 → habit change to octahedron-2.

The core shows the most intricate pattern (Figure 3b–d) produced by unstable conditions during early crystal growth. Judging by the contours of the growth zones, the crystal was growing as an octahedron, but the growth layers shifted, one relative to another, while the nucleation center remained perfectly homogeneous. The inclusion of chromite is located in the center of the homogeneous zone, but is not expressed in any way in the cathodoluminescence pattern, as well as in the pattern of structural etching (Figure 4).

The vertices of the central zone at the top left and bottom right are intersected by uneven surfaces that intersect rectilinear growth zones. These surfaces can be interpreted as mechanical damage (chips) (Figure 3c,d).

The transition zone records a dramatic change in the growth conditions, when a cuboid began developing after the octahedron (Figure 3e–g). The final habit corresponds to a typical cuboid with convex faces and obtuse vertices truncated by small, octahedral faces (Figure 3e–g). The cuboid is homogeneous and free from fibers, sectors, or growth layers, though consists of two similar growth zones (Figure 3e). The faces of the octahedron are developed at the vertices of the cuboid zone (Figure 3f,g). With this growth mechanism, each following octahedral layer has a smaller surface area than the previous one, and the cuboid surfaces acquire a complex microtopography, while the octahedral faces make the cuboid vertices obtuse.

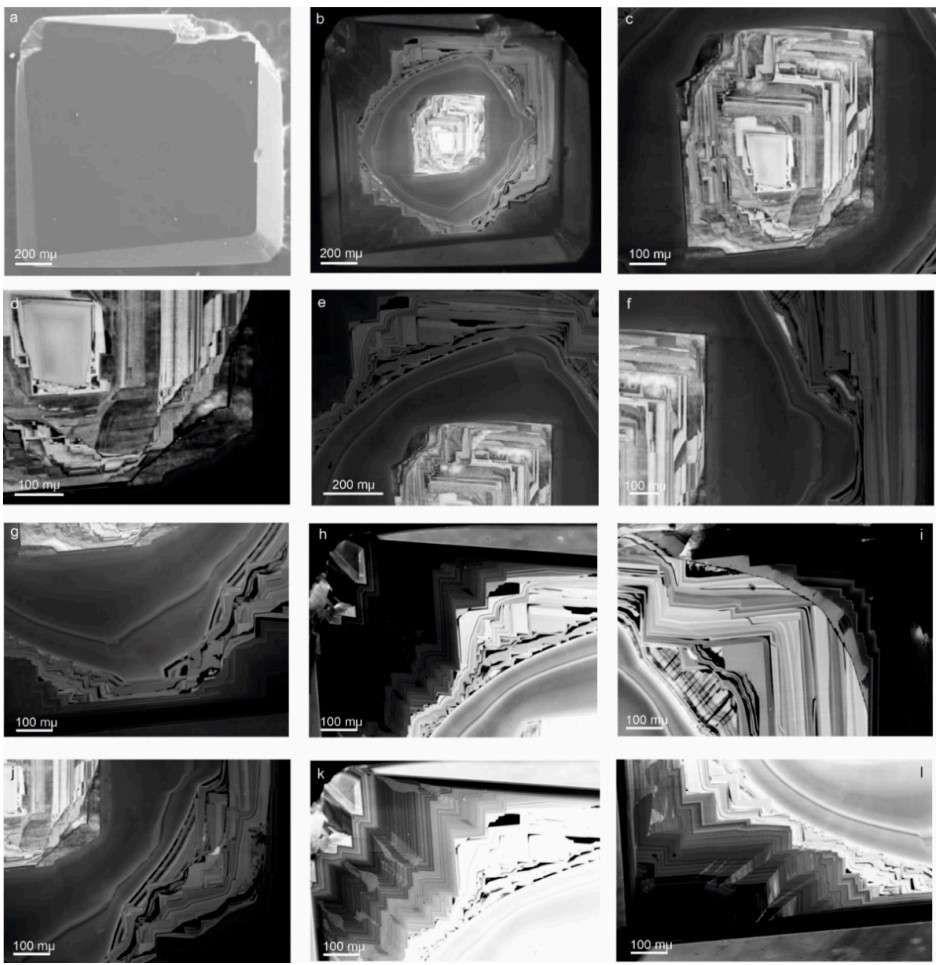

**Figure 3.** CL images of diamond plate 4007. (**a**): general view; (**b–d**): core; (**e–h**): transition zone; (**i–l**): transition zone and rim.

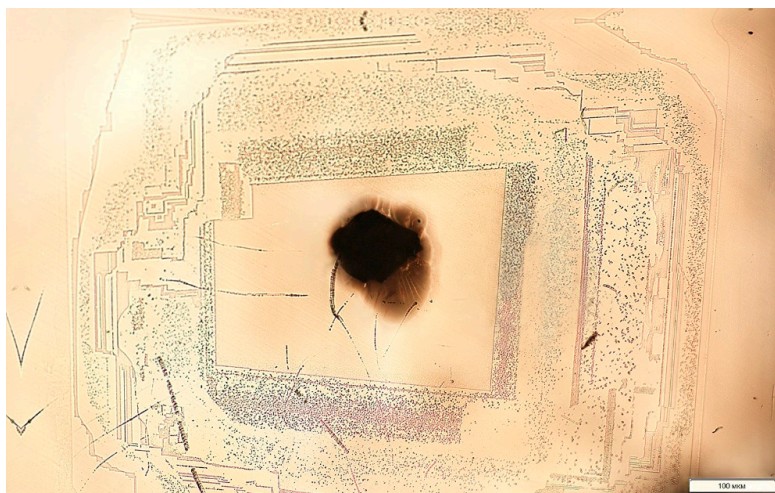

**Figure 4.** Inclusion of chromite in the central homogeneous zone against the background of etched sculptures (optical micrograph).

The rim consists of two subzones: a light inner subzone, and a dark outer one. Before the formation of the inner subzone, the cuboid apparently underwent minor dissolution, which produced pits on the surface of the cuboid zone (Figure 3f,j). Initially, this subzone has a jagged, micropyramidal character over the convex surface of the cuboid zone; the

faces of the micropyramids correspond to an octahedron. In the outer subzone, growth had a columnar character. Above the vertices of the cuboid, growth continued with octahedral layers (Figure 3f,g). It is clearly seen that growth over the convex surface of the cuboid is the most rapid and blocky, while over the vertices of the cuboid, growth is slow and layered (Figure 3f,g). The convex cuboid surfaces were quickly overgrown with columns, while the vertices became slowly overgrown, layer by layer, and the crystal finally acquired an octahedral habit. After the formation of its inner subzone, the crystal underwent partial dissolution (Figure 3i), then octahedral growth resumed following the inherited columnar patterns, i.e., it was likewise columnar (Figure 3k,l). Eventually, the cuboid crystal transformed to a flat-faced octahedron.

### 3.2. Selective Etching

The image of the etched diamond plate (Figure 5) highlights the same zones as the CL image, with flat-bottomed center-ward pits on the crystal surface (Figure 4) that have symmetry corresponding to that of the growth sectors. In the absence of etch channels along dislocations, the pits most likely represent point defects, which are also responsible for the CL patterns, while the density of the pits corresponds to that of the defects. The density of the defects is greatest along the periphery of the core, while its center is almost free from defects, as in the CL image (Figures 4 and 5a,b). Defects are few in the transition zone (corresponding to the cuboid growth), which is free from growth sectors or fibers, while growth zones appear only at the final stage (Figure 5c,d).

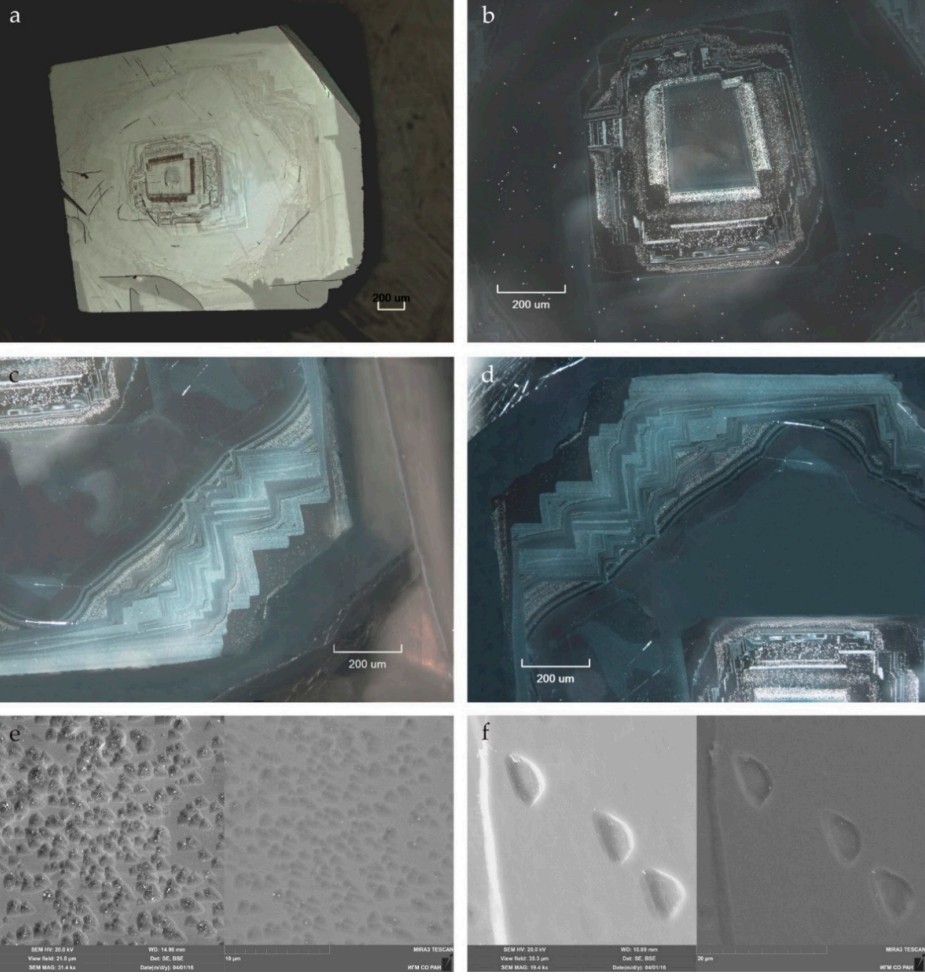

**Figure 5.** Selective etching of diamond crystal 4007. (**a**): general view; (**b**–**d**): details of etching pattern; (**e**,**f**): SEM image of pits.

At the next stage, when growth conditions changed and the cuboid transformed into an octahedron, the crystal acquired numerous defects and prominent growth zones; however, the pits were very small (Figure 5c,d). The CL and etching patterns are almost identical and must reveal the same defects.

### 3.3. FTIR Mapping

According to IR data, nitrogen in the sample is present in the form of A defects (two nitrogen atoms in adjacent carbon-substituting positions), and B-defects (four nitrogen atoms in a carbon-substituting position plus a vacancy), which allows us to attribute the crystal to the IaAB type. The results of studying the distribution of concentrations of these forms of nitrogen, as well as the intensity of the platelites (B') peak, are shown in Figure 6.

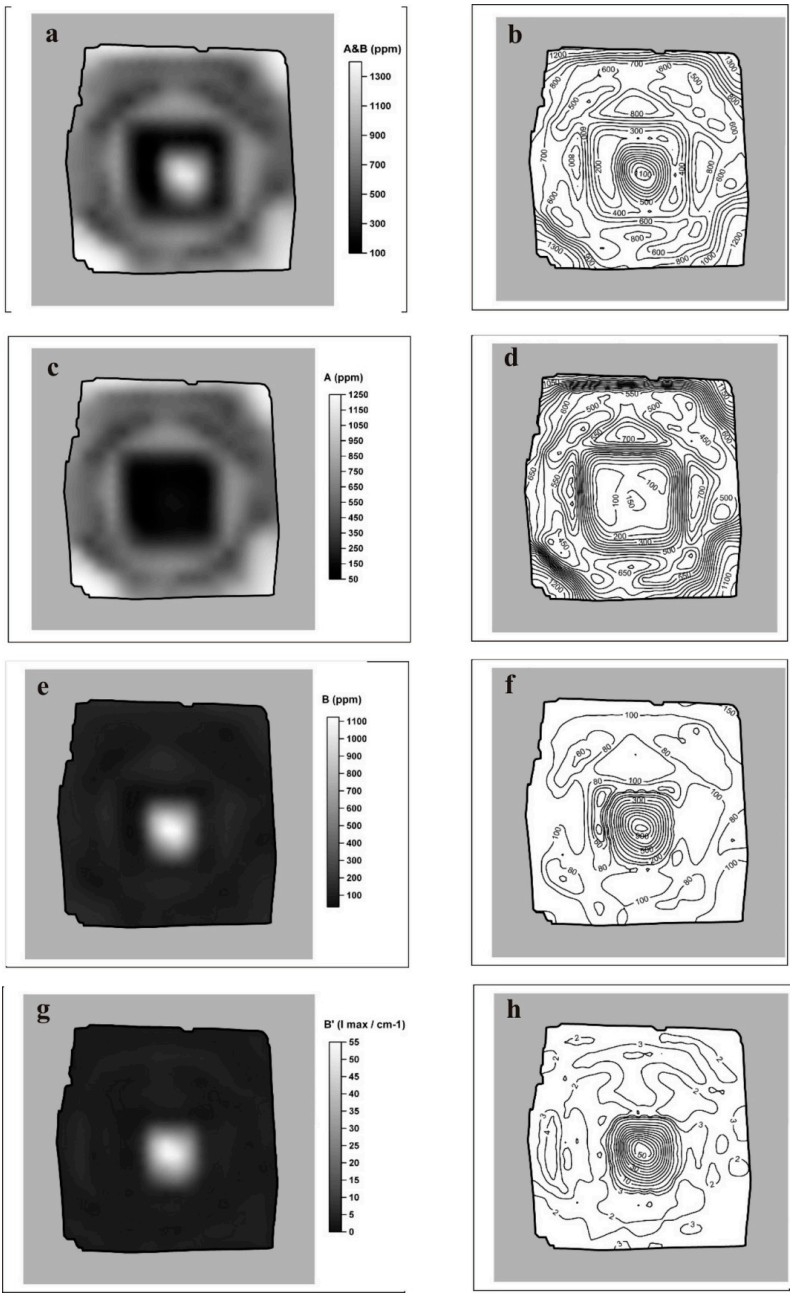

**Figure 6.** Results of IR mapping of diamond plate 4007. Spatial distributions are shown: (**a**,**b**): total nitrogen, calculated as the sum of concentrations A and B of nitrogen defects; (**c**,**d**): nitrogen in the A form; (**e**,**f**): nitrogen in the B form; (**g**,**h**): the intensity of the platelets peak B'.

The total nitrogen content (Figure 6a,b), defined as the sum of nitrogen concentrations in the A-form (Figure 6c,d) and B-form (Figure 6e,f), varies in the sample from 115 to 1400 ppm. At the same time, its distribution over the crystal is very nonuniform. In the central area of the crystal corresponding to octahedral growth, two subzones are distinguished (visible both in cathodoluminescence and structural etching). In the inner subzone, the maximum content of total nitrogen is observed up to 1280 ppm, with the maximum degree of its aggregation into B centers reaching 85%–87%. In the outer subzone, these parameters begin to noticeably decrease. In the middle region (transition zone) of the crystal, corresponding to the growth of the cuboid, the total nitrogen content reaches 120–500 ppm, and the degree of its aggregation also continues to decrease nitrogen, which is already represented mainly by the A-form. At the same time, in the outer peripheral region of the crystal, corresponding to growth along the octahedron, the nitrogen content again increases and reaches 1200–1400 ppm, but the degree of nitrogen aggregation into B centers does not exceed 10%–15%. Thus, the central and peripheral regions of the sample are most enriched in nitrogen, with a general decrease in the degree of its aggregation into the B form from the center to the outer zones of the crystal.

The platelets peak associated with lamellar precipitates along the planes of the cube, which are a by-product of the formation of nitrogen in the B-form [27], also manifests itself inhomogeneously in the sample (Figure 6g,h). Its height, the highest level of which is 53 cm$^{-1}$ (integral intensity 950 cm$^{-2}$), is observed in the central region of the crystal with a high degree of nitrogen aggregation, decreasing to minimum values as the degree of aggregation decreases in the middle and peripheral regions of the crystal. For the central, highly aggregated region of the crystal, there is a linear correlation ($R^2 = 0.997$) between the intensity of the platelet peak and absorption associated with nitrogen in the B form, which makes it possible to characterize this region as "regular" i.e., formed without platelete degradation [27]. At the same time, in the transitional cuboid region, and especially in the outer parts of the crystal with a low degree of nitrogen impurity aggregation, deviations from this correlation are observed. Their nature, taking into account the variation in the position and shape of the platelete peak, as well as the presence of indicative peaks at 1525 and 1550 cm$^{-1}$, makes it possible to classify these zones as "subregular" [28]. It is supposed that insufficient development of platelets occurred in this area due to a decrease in temperature.

Of the additional features in the IR spectra, there is a weakly pronounced absorption peak at 3107 cm$^{-1}$, associated with hydrogen-containing defects (with an integral intensity of up to 20–25 cm$^{-2}$, locally up to 50–55 cm$^{-2}$). Also, in some spectra of the central region of the sample, directly near the visible inclusion, a weak peak at 1430 cm$^{-1}$ (carbonates?) is determined.

In general, we note that the described features of the IR spectra, which reflect the regularities in the distribution of the main structural impurities, in particular, nitrogen defects, are in good agreement with the cathodoluminescence and etching patterns described above, which also reflect the features of the defect-impurity structure of the sample studied.

### 3.4. Photoluminescence

The transmission and photoluminescence (PL) spectra (Figure 7a) were measured in three zones: the strongly luminescent center of the core that grew by the octahedral mechanism (1); a weakly luminescent transition zone (2); and the rim (3), which was formed by cuboid and octahedral growth, respectively.

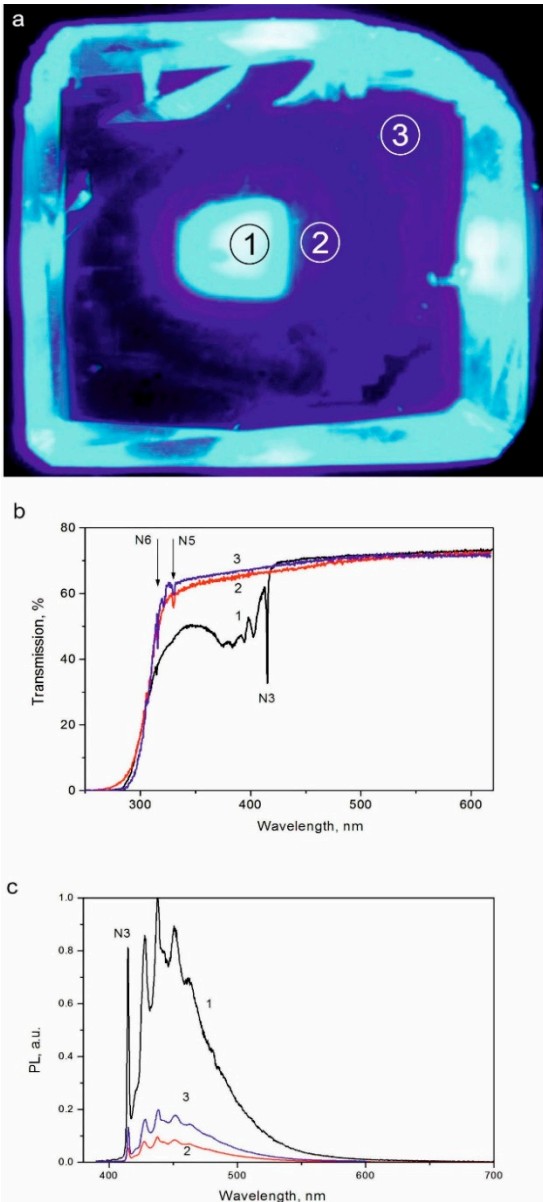

**Figure 7.** Diamond plate 4007: (**a**): Pl image, UV excitation; 1, 2, and 3, are zones where transmission and PL spectra were collected. See the chromite inclusion at the center of the strongly luminescent zone; (**b**): Transmission spectra measured at three points, shown in Figure 7a: strongly luminescent center of octahedral core (1); weakly luminescent cuboid transition zone (2); and octahedral rim (3); (**c**): PL spectra obtained with 350 nm excitation at 80 K in points 1, 2, and 3, respectively.

The crystal becomes roughly transparent from 290 nm, which is common to diamonds of Ia type, where nitrogen is aggregated in pairs (A centers) or in more complex centers. Narrow lines N5 (329.6 nm or 3.762 eV) and N6 (315.6 nm or 3.928 eV) in spectra 2 and 3 are typical of IaA-type natural diamonds (Figure 7b).

The spectrum of the strongly luminescent octahedral core (point 1) includes an electron vibration system, N3, with a narrow no-phonon line of 415.2 nm or 2.985 eV, and a broad band within the 350–415 nm range at the center (Figure 7c). The N3 center is the most widespread N defect in natural IaA, IaB, and IaAB-type diamonds [29], which was modeled as a system of three nitrogen and a vacancy ($N_3V$). The intensity of the zero-phonon line commonly does not exceed 3 cm$^{-1}$ and estimates of the concentration of these centers, taking into account the strength of the oscillator and the refractive index of diamond $n_\lambda$

from [30,31], give $4 \times 10^{16}$ cm$^{-3}$. In our case, in the core zone, the intensity of zero-phonon line is 12 cm$^{-1}$ and the defect concentration approaches $10^{17}$ cm$^{-3}$.

The PL spectra for points 2 and 3 are similar in shape to N3, but are at times weaker than in the core. In general, the blue PL pattern in Figure 6, matches the signals from nitrogen B centers in the FTIR maps.

### 3.5. Birefringence

The diamond plate we analyzed likewise shows anomalous birefringence, associated with the distribution of defects in the core, transition zone, and rim, respectively, which were formed by octahedral, cuboid, and octahedral growth (Figure 8). The chromite inclusion in the core center produces a local field of anomalous birefringence, associated with decompression and the thermal expansion difference between diamond and chromite, and the related stress [32].

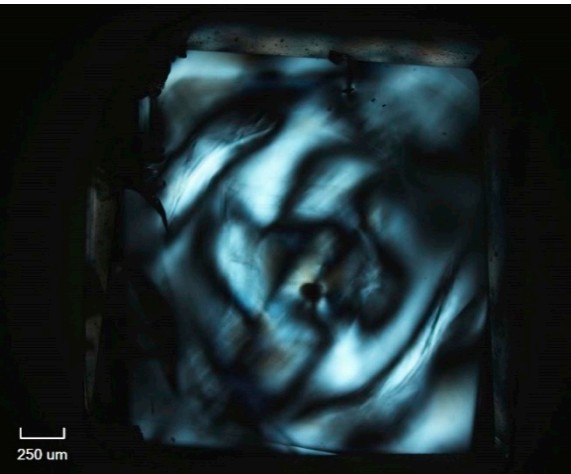

**Figure 8.** Anomalous birefringence of diamond crystal 4007.

### 3.6. Raman Mapping of Stress in Diamond around the Chromite Inclusion

The stressed chromite inclusion in diamond was analyzed by Raman mapping. The inclusion size is maximum ~100 µm across.

The x-y stress field around the inclusion was reconstructed from the position of the diamond band (1332 cm$^{-1}$ at 1 bar) at the constant depth z corresponding to the distance (z ~80 µm) from the inclusion center to the free surface of the diamond plate. The Raman signal between 1100 and 1500 cm$^{-1}$ bands were acquired at each point to a spatial resolution of 5 µm ($5 \times 5 \times 5$ µm$^3$ in volume). The scanned x-y field is $220 \times 220$ µm$^2$. Color in the Raman image (Figure 9a) refers to the red-blue shifts of the wavenumber of the diamond band, that is equal to 1332.0 cm$^{-1}$ (dark blue) in non-stressed diamond (dominant part). The wavenumber varies over the specimen from 1331.5 cm$^{-1}$ (black) to 1333.34 cm$^{-1}$ (red). The red and black colors correspond, respectively, to the maximum positive (1.34 cm$^{-1}$) and negative ($-0.5$ cm$^{-1}$) shifts. The black color area in the upper part of picture, exhibiting the 1331.5 cm$^{-1}$ band, probably has a larger content of N-impurities, causing some heating of the sample in focal point of laser beam.

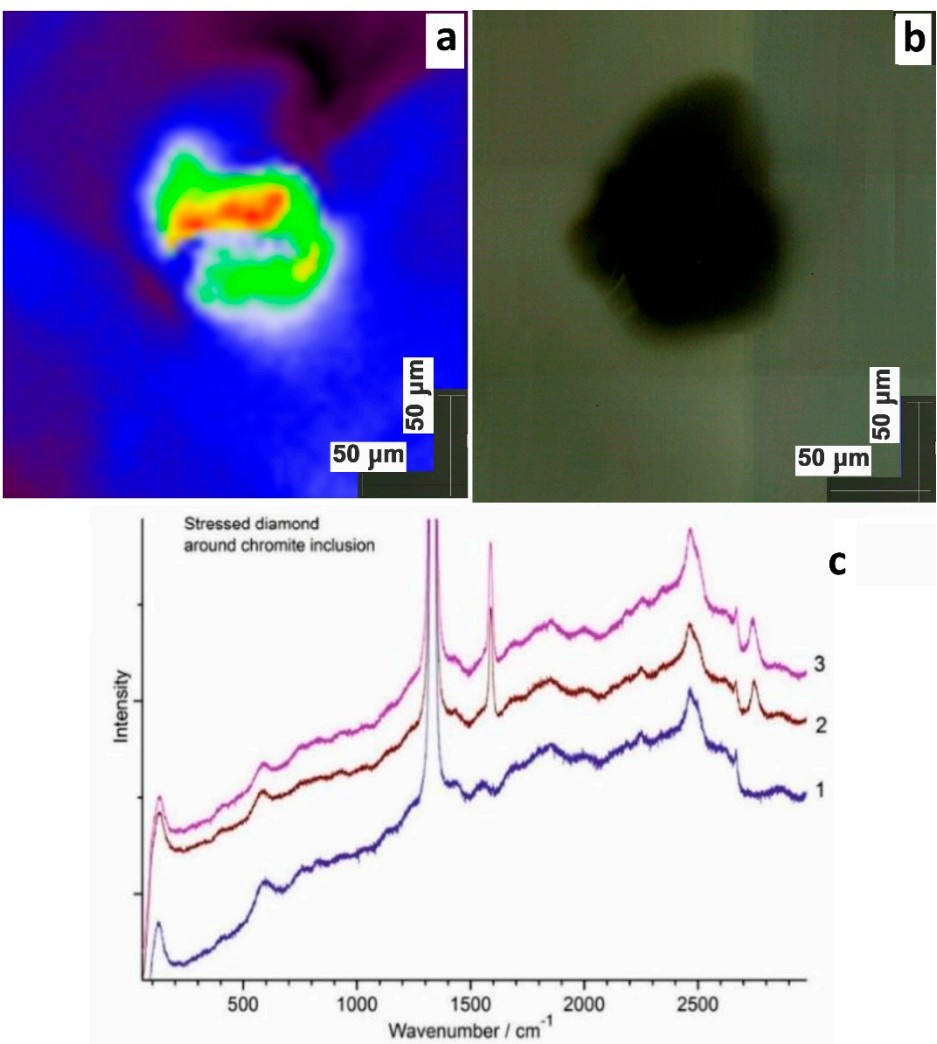

**Figure 9.** (**a**): Raman map of stress around the chromite inclusion in diamond plate 4007; (**b**): photo of this chromite inclusion in the same scale. X and Y bars are equal to 50 μm; (**c**): representative Raman spectra of diamond around the chromite inclusion in this plate 4007. The inclusion size is maximum ~100 μm across.

The Raman spectra of the stressed regions around the inclusion (Figure 9b) reveal the presence of graphite in a narrow zone near the inclusion surface (Figure 4). Graphite is identified reliably according to first- and second-order bands at 1583 cm$^{-1}$ and ~2725 cm$^{-1}$, respectively (spectra 2 and 3 in Figure 9b). All spectra include a strong low-frequency band of as yet unknown origin at 140 cm$^{-1}$.

### 3.7. LA-ICP-MS

The concentrations of selected, detectable impurity elements were analyzed in four growth zones (1–4) of the diamond plate, numbered from core to rim, in individual spots, and as average over the zones (Table 1). Concentrations are the highest for B and Cr: 0.5–1.75 ppm and 0.7 to 1 ppm, respectively, while Sr, Y, and Zr, are very low (0.5 to 31 ppb). The growth zones differ in composition (Table 1; Figures 10 and 11), with the highest Cr and lowest B and Ti in the core (zone 1); Ni, Cu, Mg, Y, and Zr enrichment, and Al depletion, relative to other zones in Zone 2; highest Ti and lowest Cr contents in Zone 3; low Ni in Zone 4, where Ti and Cr contents are comparable to those in Zone 2, while Mg and Al contents are about the same as those of Zone 3. Averaged totals of all selected elements are lowest in the core (2.3 ppm), and 3.26–3.32 ppm in all other zones. The contents of B and Ni (Figure 10) plot different fields for zones 1 and 2, but overlap for zones 3 and 4

(Figure 10). Ti and Cr (Figure 10) of zones 1 and 3 are in negative correlation, while those of zones 2 and 4 occupy an intermediate position.

**Table 1.** Average element concentrations (ppm) in four growth zones of diamond.

| Element | B | Mg | Al | Ti | Cr | Ni | Cu | Zn | Sr | Y | Zr | La | Sum |
|---------|------|------|------|------|------|------|------|------|------|------|------|------|------|
| Zone 1 | 0.578 | 0.278 | 0.396 | 0.015 | 0.991 | 0.0343 | 0.0113 | 0.0186 | 0.0012 | 0.0006 | 0.0031 | 0.0002 | 2.33 |
| Zone 2 | 1.541 | 0.308 | 0.344 | 0.032 | 0.826 | 0.1605 | 0.0370 | 0.0557 | 0.0016 | 0.0023 | 0.0112 | 0.0003 | 3.32 |
| Zone 3 | 1.752 | 0.263 | 0.391 | 0.067 | 0.723 | 0.0161 | 0.0136 | 0.0341 | 0.0012 | 0.0003 | 0.0011 | 0.0001 | 3.26 |
| Zone 4 | 1.676 | 0.273 | 0.380 | 0.038 | 0.836 | 0.0081 | 0.0203 | 0.0535 | 0.0005 | 0.0012 | 0.0007 | 0.0004 | 3.29 |

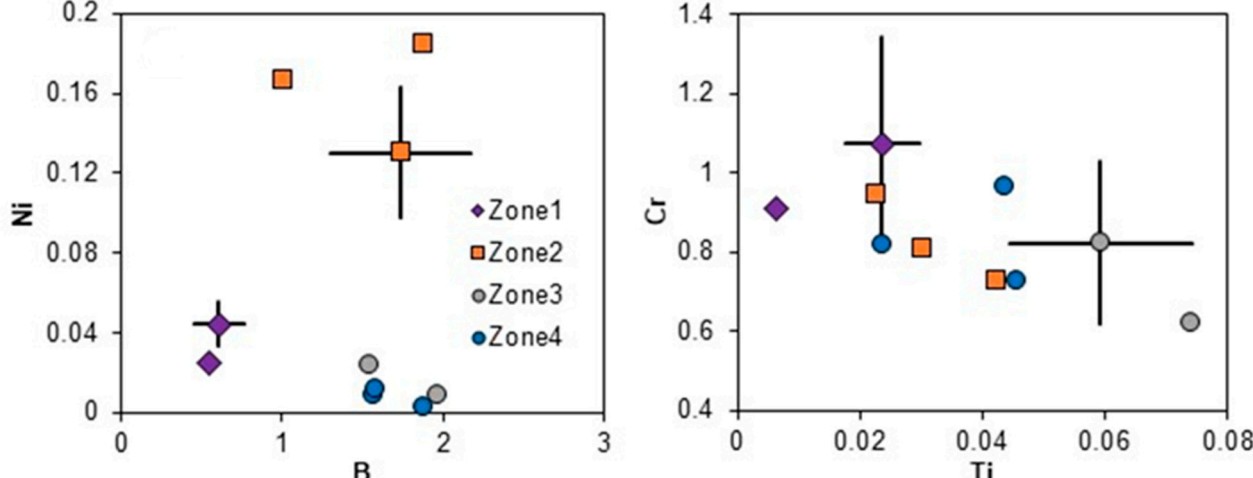

**Figure 10.** B-Ni and Ti-Cr diagrams (ppm) in four growth zones of diamond. Error bar corresponding to 25% of relative deviation added to selected points.

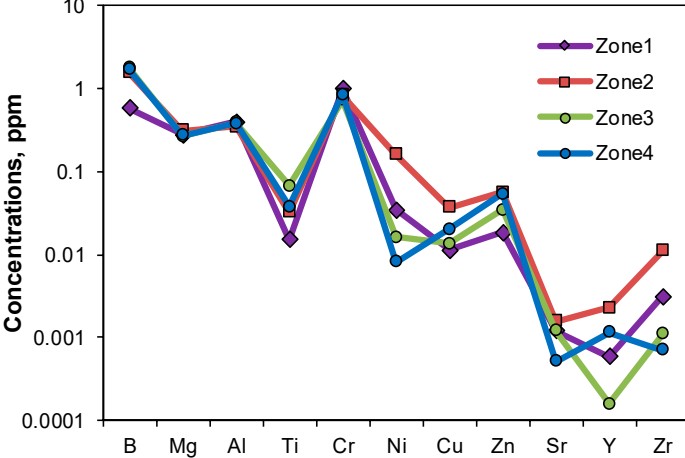

**Figure 11.** Average element contents (ppm) in four growth zones of diamond.

## 4. Discussion

The diamond crystal we studied exhibits well-pronounced zoning typical of natural kimberlitic diamonds, with a core, a rim, and a transition zone. This zoning was observed in all data, except for Raman spectroscopy, which only revealed stress in the diamond around the chromite inclusion. Weaker stress over the specimen is poorly resolvable in Raman spectra, but is detectable in anomalous birefringence. The zoning, with defects of different morphologies and types in the three zones, records variations in the conditions of crystal growth at the respective stages.

The core of the crystal has the most complicated structure (Figures 3 and 4), highlighted in CL and selective etching patterns. It has a homogeneous central subzone, with the greatest concentration of nitrogen B centers and, correspondingly, with the highest N aggregation and density of platelets ("regular" subzone). The regular subzone is surrounded by intricately alternating thin layers, with curved boundaries truncating the layered structure, which record repeatedly interrupted growth in a nitrogen-rich medium under unstable conditions.

Another type of irregular outline that violates rectilinear zoning (Figures 3, 4 and 5c,d), indicates mechanical cleavage at the top of the growing crystal. Such damage associated with breaks in growth is often seen on the inner zones of growing crystals, for example, on a crystal from the Snap Lake deposit (Canada) (Figure 12).

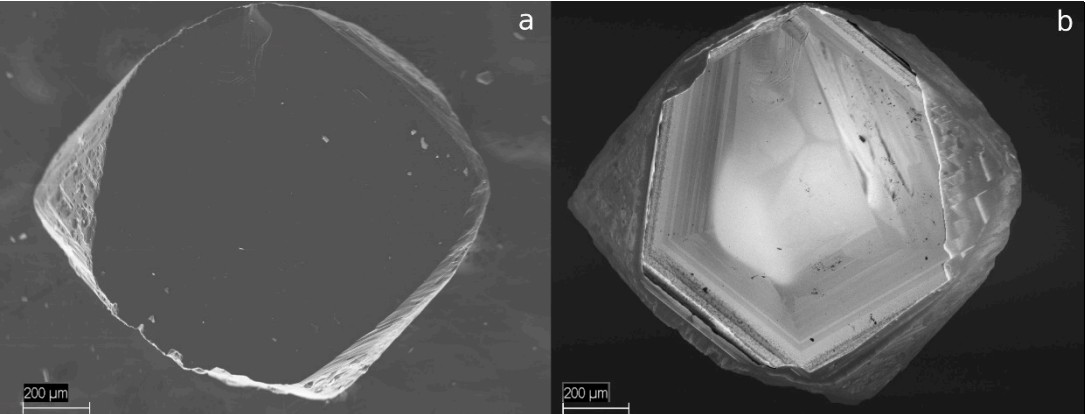

**Figure 12.** Cathodoluminescence pattern of a Snap Lake diamond showing mechanical damage on the right and top left, which later regenerated. (**a**): the contour of the diamond plate; (**b**): picture of cathodoluminescence.

Mechanical cleavage at the top of the studied diamond's central zone shows that after the completion of its growth, the crystal exhibited brittle properties, which required a temperature below the plastic flow temperature, and a short exposure time, at which the brittle properties of the crystal were maximally manifested.

The chromite inclusion, if originally present in the core, may have interfered with the normal crystal growth. It remains unclear whether the chromite inclusion was a seed for diamond nucleation (by unknown mechanisms), or a zone of abundant defects with numerous submicron and nanometer inclusions that coalesced into a single defect as a result of annealing [33,34]. The latter option appears more feasible; coalescence of submicron or smaller inclusions is known from other mantle minerals, such as pyrope, Mg ilmenite, or chromite [35]. If diamond is treated as having its growth and transformation patterns the same as in any other mineral, its crystal structure can be expected to change by diffusion, which is a recognized mechanism of nitrogen aggregation. Thus, thermodynamically unstable zones with a high density of defects are prone to ordering, while less defective zones may remain metastable. This inclusion now forms an anomalous stress field, which is clearly visible in anomalous birefringence, and which we studied by Raman spectroscopy.

The octahedral core grades directly into a homogeneous transition zone, ~400–500 μm across. Judging by the contours of the zone, the crystal growth mechanism changed from octahedral to cuboid at that stage. The growth of the cuboid was carried out by octahedral layers, this can be seen from the faces of the octahedron at the vertices of the cuboid zone (Figure 4). However, in the process of growth, the vertical growth rate of the octahedron faces was greater than the tangential velocity, i.e., new octahedral growth layers appeared before the end of the lower layers' completed growth. Therefore, in place of the face of the octahedron, a layered pyramid was formed, consisting of decreasing octahedral growth layers. In this case, the growth layers were rotated by 60° relative to the normal face of the

octahedron. As a result, a cuboid shape developed, in which the cuboid surfaces have a complex micromorphology, and do not correspond to the real crystallographic face of the cube. This growth mechanism is characteristic of diamond cuboids. As an example, we can show diamond cuboids from the Kumdy-Kol deposit (Kazakhstan). In these diamonds, the growth layers had the shape of a ditrigon; therefore, in addition to the surfaces of the cuboid, the surfaces of the dodecahedroid were also formed, and the faces of the octahedron blunt the vertices of the cuboid (Figure 13).

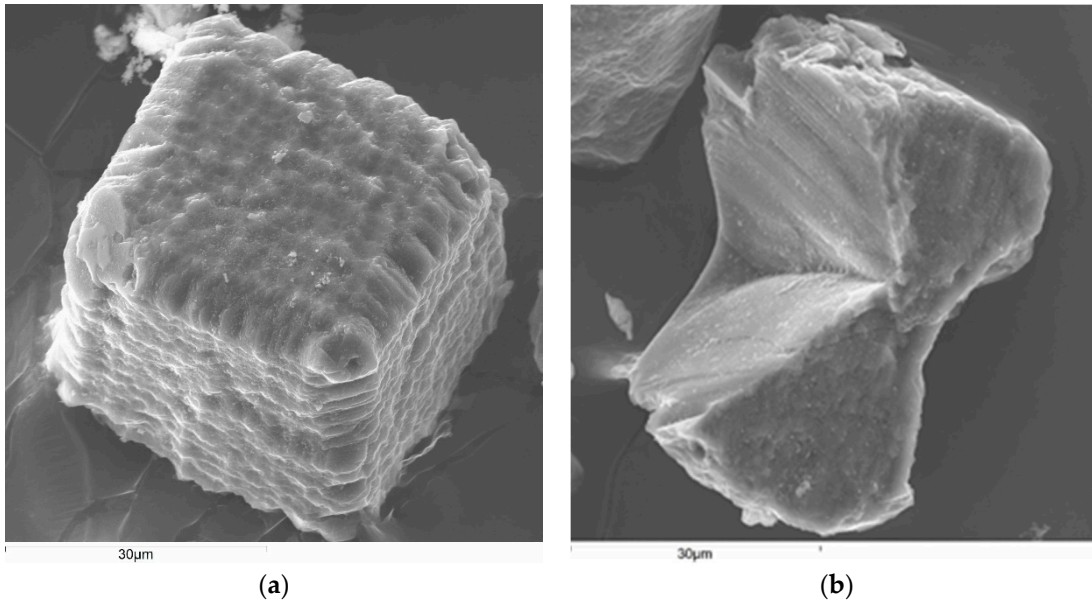

(**a**) (**b**)

**Figure 13.** Diamond cuboids from the Kumdy-Kol deposit (Northern Kazakhstan). (**a**): whole crystal; (**b**): crystal fragment, with evident octahedral growth layers.

Thus, the transition zone was growing with the cuboid mechanism, but with thin octahedral growth layers and without developing typical fibrous textures common to cuboid growth. Nitrogen in the transition zone has a lower total content and aggregation degree than in the core, and mostly occurs as A centers.

If the growth of a diamond stopped at a cuboid, then we would have a "coated diamond". However, the period of growth along the cube was replaced by growth along the octahedron, which again indicates a change in the conditions of crystallization. Moreover, the surface of the cuboid was apparently subjected to slight dissolution, which can be seen from the etching pits crossing the outer thin contours of the growth layers along the cuboid surfaces (Figures 3 and 5).

Initially, inner subzone has a jagged, micropyramidal character over the convex surface of the cuboid zone; the walls of the micropyramids correspond to an octahedron. In the outer subzone, growth had a columnar character. Above the vertices of the cuboid, growth continued with octahedral layers (Figure 3f,g). It is clearly seen that growth over the convex surface of the cuboid is the most rapid and blocky, while over the vertices of the cuboid, growth is slow and layered. Crystal growth ends with the formation of an octahedron.

The same zoning appears in the FTIR spectroscopy results, which allow us to estimate the total contents and speciation of the nitrogen impurity. The total N contents are high in the core and rim zones formed by octahedral growth, but relatively low in the transition zone that grew by the cuboid mechanism. The nitrogen aggregation is high only in the core but is lower in the transition zone, as well as in the rim, where most of nitrogen occurs at A centers. Thus, the aggregation of nitrogen decreases rim-ward, and poorly correlates with the total N contents.

Diamond forms in the mantle at high pressures and temperatures. Decompression and quenching of the erupting kimberlite magma which carries diamond to the surface, as

well as the presence of nitrogen defects, cause stress to the crystals. The stress appears as anomalous birefringence and is detectable in polarized light. Anomalous birefringence, as a reflection of structural inhomogeneity, has been studied for a long time [9–15]. Anomalous birefringence may result from: (1) impurities and defects in zones and sectors; (2) external forces, e.g., grain collisions and wear during formation of placers; (3) plastic deformation; and (4) inclusions [10]. The diamond plate we analyzed likewise shows anomalous birefringence associated with the distribution of defects (cause 1) in the core, transition zone, and rim, which were formed, by octahedral, cuboid, and octahedral growth, respectively (Figure 8). The chromite inclusion in the core center produces a local field of anomalous birefringence, associated with thermal expansion difference between diamond and chromite and the related stress (cause 4).

Raman spectroscopy also shows signs of mechanical stress [36–39], in this case around the inclusion of chromite. The diamond stress field around the inclusion comprises radial and azimuthal components, the former being more prominent than the latter (Figure 9a). Strong azimuthal stress may be associated with the non-spherical shape of the inclusion, which induces anisotropic stress at corners, as well as with difference in compressibility along (111) and (001) axes, with shifts of 2.2 cm$^{-1}$/GPa and 0.7 cm$^{-1}$/GPa, respectively [37]. The azimuthal stress may be due to non-orthogonality of the two directions (Figure 9a). Additional complex mechanic stress may arise around cracks in diamond. The interior pressure in the chromite inclusion was estimated as ~0.8 GPa.

In general, the environment changed four times during the growth of crystal 4007 and caused respective changes to the growth mechanism, as well as to the concentration and aggregation of nitrogen. The most defective core presumably became ordered by annealing under high temperature during the long growth, which produced a homogeneous zone in the center of the core. The outer contours of the zones bear traces of dissolution, indicating breaks in growth.

The zoning appeared in the data of all methods, except for Raman spectroscopy, and was especially prominent in CL and etching data. The similarity of the etching and CL patterns highlights their relation to the same structure defects. Judging by the relatively flat bottoms of the pits (Figure 5) and the absence of etch channels, the defects were likely point-like and associated with nitrogen impurity, rather than being linear dislocations. This inference is consistent with the FTIR and PL data.

Previously, it was found that N-poor type IIa diamonds are more resistant to etching than type I diamonds, which have a ~330 times higher pitting density [40]. Thus, the resistance of diamonds to selective etching largely depends on the density of nitrogen defects. Meanwhile, our results show that type Ia nitrogen-rich diamonds, like the crystal we analyzed, can resist etching due to structure ordering upon annealing, which is thus another stability factor.

Multi-stage growth is characteristic of many diamonds. It is a very difficult task to restore the conditions for each stage of growth, therefore, a simplified approach is usually used. This consists of correlating a diamond as a monogenic formation with a certain mineral paragenesis (eclogitic or peridotite), which is considered to be the parent for this diamond. The conducted study shows that for a more complete understanding of the genesis of diamond, it is necessary to look for ways to restore environmental conditions at each stage of diamond formation, and correlate them with the development of the lithosphere. Figure 14 shows the trend in the estimated average temperatures of mantle residense (the period of formation and post-crystallization annealing) for different zones of the diamond 4007: from 1140–1200 °C in the central region with a high degree of nitrogen aggregation to 1060–1125 °C in the peripheral low-aggregation zone.

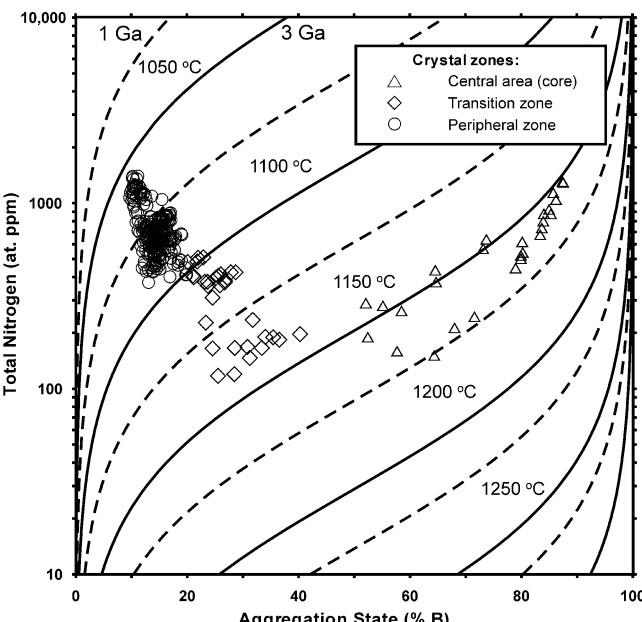

**Figure 14.** Plot of the degree of nitrogen aggregation (% B) in diamond 4007 versus its total nitrogen content, with added grid of isotherms of mantle residence for 1 and 3 Ga (calculated using 2nd order kinetic equation and nitrogen aggregation parameters from [41]). The central area of the crystal is "regular", nitrogen aggregation varies from 50% and above, the points are located in the isotherm range 1140–1200 °C. Transition zone of the crystal show deviations from "regularity" with a shift of points towards lower temperatures, nitrogen aggregation decreases to 20%. The peripheral part of the crystal is "subregular", nitrogen aggregation is 10%–20%, the points are localized in the region of isotherms with temperatures of 1060–1125 °C.

It should be noted that such a level and the decreasing character of temperature changes are common for this type of diamond with a "regular" core and a "subregular" peripheral part of the crystal [28]. The analyzed diamond crystal nucleated at quite high P-T parameters (octahedral core), then grew by the cuboid mechanism at a lower temperature, and finally the octahedral rim developed at a still lower temperature. During gaps in the growth, the central part of the crystal was subjected to mechanical damage, and the cuboid zone was partially dissolved.

Diamond is the only mineral that can record the evolution of the respective region of the lithospheric mantle in its growth history, which can be reconstructed from the crystal structure. It is likely that the temperature trend during crystal growth is a consequence of mantle dynamics in the process of diamond formation, and some speculations in this regard can be found in the review [42].

## 5. Conclusions

The studied diamond shows four discrete stages of growth under different physico-chemical conditions. The revealed structural inhomogeneities, with the exception of the anomalous stress field around the chromite inclusion, are associated with the distribution and forms of structural nitrogen impurity. All kinds of study, except for Raman spectroscopy, have shown this.

The conducted studies show that for a more complete understanding of the genesis of diamond, it is necessary to look for ways to restore environmental conditions at each stage of diamond formation, and correlate them with the development of the lithosphere. Thus, the history of diamond growth simultaneously shows the history of the fragment of the lithosphere in which the diamond was located.

**Author Contributions:** Conceptualization, V.A., S.U., Y.B., V.S., A.L., A.Y., S.G., A.A.; methodology, V.A., S.U., Y.B., A.L., A.Y., S.G., A.A.; investigation, V.A., S.U., Y.B., V.S., A.L., A.Y., S.G., A.A., O.I.; writing—original draft preparation, V.A., S.U., Y.B., A.Y., S.G., A.A.; writing—review and editing, V.A., S.U. All authors have read and agreed to the published version of the manuscript.

**Funding:** This research was funded by the state assignments of IGM SB RAS and DPMGI SB RAS.

**Data Availability Statement:** Not applicable.

**Acknowledgments:** We are very grateful to the Analytical Center for Multielemental and Isotope Research SB RAS (Novosibirsk, Russia), where analytical studies were carried out. We thank Kirill Ponkratov for Raman mapping of the diamond sample, carried out at the Renishaw Company.

**Conflicts of Interest:** The authors declare no conflict of interest.

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
