# Peer review of "Growth Story of One Diamond: A Window to the Lithospheric Mantle"

_minerals, doi:10.3390/min12081048_

Round 1

Reviewer 1 Report

The manuscript reports on physical and chemical characteristic of a single diamond with a chromite inclusion in the center and interprets a growth history of the diamond in terms of temperature, stress and a growth mechanism (cubic vs octahedral). The paper is well written, superbly translated, concise and to the point. I recommend accepting the manuscript after a major revision. Main points that need to be addressed are listed below, while comments keyed to specific lines and paragraphs are inserted as marginal notes on the attached file.

·       The manuscript would benefit from using more recent references, and I suggested several modern reviews that can be quoted.

·       Data on trace elements were acquired but not used. We do not even know if the contrast in elemental concentrations between zones is significant (i.e. exceeds the analytical errors) or not. Could the concentrations be used to infer the diamond parageneses in different zones or to confirm the peridotitic affinity of the central zone?

·       The authors should expand interpretation of the temperature of diamond formation to get insights into mantle processes.

Hope my comments improve the quality of the manuscript. I am looking forward to see the study published.

Author Response

P.38. Thank you for your valuable comment. I've added a link.

pp.54-56. Moved this to the discussion.

pp.64-65. Thanks, I added modern research methods.

pp.135-137. Added the required information.

P.143. Corrected.

pp.151-152. Corrected.

P.196. IR mapping is based on the cooperative processing of a large number of spectra recorded at different points in the sample. There can be hundreds or even thousands of such points, which makes it inconvenient to present data in the form of tables. In our case, such a table would also consist of several hundred rows and would take up several pages. Therefore, the authors present the data obtained in a compact, but at the same time informative, graphical form (Fig. 4), as in other publications used this method (see for instance: Contrib. Mineral. Petrol., 2012, V.164, Issue 6, P.1011-1025; Diam. and Rel.Mater, 2016, v.69, p.8-12; Contrib.Mineral.Petrol., 2018, p.173:39; etc.) .

P.250-252. Corrected

pp.276-282. Thank you, moved to the discussion

P.286. I agree with you. Added a link to the work you recommended.

P.305. Added the required information.

pp.308-318. Moved this to the discussion.

pp.373-377. Thank you for your comment. I have described in more detail.

S.432-440. I have corrected the text.

  1. 455-462. The reviewer suggested adding a discussion of the results obtained from the diamond sample as applied to mantle processes. However, this recommendation goes beyond the scope of the mineralogical tasks set in this work. In addition, data from one sample is clearly not enough for reasonable assumptions of a geodynamic nature. Therefore, the article was simply added a reference to a more general and representative review (Nimis, 2022), where there is both a discussion of a large number of estimates of the P,T-parameters of diamond formation available in the literature, and some speculations about diamond formation in connection with processes in the mantle.

P.468. Added an explanation.

Thanks for the strict and good review. The review really helped improve the article.

Reviewer 2 Report

The article presents the results of a combined study on a diamond sampled at the XXIII CPSU Congress kimberlite at Mirny city. The article needs minor revisions to be published in an impacted journal like Minerals.

1.       Lines 16, 72, 476. There is written “local version” as a specification of the Raman spectroscopy. What does it mean? Is it extremely necessary to write this phrase?

2.       Is it possible to spend some words about the geological setting where the diamond is sampled? Is it possible to show a map and introduce some references for it?

3.       Line 80: “The analyzed sample was cut out of a transparent colorless….”. Are there any yellowish parts in the sample?

4.       Figure 1. Is it possible to add a scale to the figure?

5.       Line 151 (also line 363): “The 151 cleavage (?) ….” Is it cleavage or not? The question mark does not give any added value to the article.

6.       Figures 2 and 4. Is it possible to elaborate the figures with a color gradient?

7.       Lines 279-282: is it possible to add more references for each anomalous birefringence case?

8.       Figure 7b. It is better to describe the spectra 1, 2, and 3 in the caption. Double-check if in the text all of these 3 spectra are described.

9.       Lines 326-328: The text shall be followed by a figure where the 4 zones are reported. It will be easier to follow the description of the ICP measurements.

10.   Is it possible to explain the Zr enrichment? Is it possible for a Zircon or fluid inclusion? Is it also possible to show a close-up of the mentioned chromite inclusion? Figure 1 shows different black areas that could be interpreted as? Inclusions? Or what? Close-up of these areas?

11.   Figure 10: Do you need to mention credits?

12.   References: they are not exhaustive. There are many masterpieces on diamonds that are not cited. You have cited many Russian articles. Although science is without frontiers, good citation work increases the value of research.

13.   The title. You have analyzed one diamond. The title should represent the dimension of the research. The current title looks like a title for a review or a title for ultimate research. Although I wish you the best of success with this research paper in order of citations and others, we are just talking about one diamond analyzed.

14.   Double-check in English, and as minor tasks, you should review the references in order to be managed according to the journal’s policies.

Although this research is interesting, the paper needs to be reviewed in some of its parts. It is accepted with minor revisions.

Author Response

  1. Corrected in accordance with the remark.
  2. Added a drawing and some geological information.
  3. There is no yellow color in the sample, the entire sample is colorless. Nitrogen in the form C, which can give a yellow color, is absent.
  4. Added a scale to the drawing.
  5. Explained in the text why we consider these surfaces to be cleavage surfaces.
  6. The form of the results output during IR mapping is determined by the capabilities of the programs used for the analysis of diamond spectra. Unfortunately, the current version of the program we used do not provides output of color images. However, in any case, the presented data contain all the key information necessary for the article, both for a general visual representation of the spatial distribution of nitrogen defects in the sample (greyscale images in Fig. 7 a, c, e, g), and more accurate quantitative information on defects content (images with isolines in Fig.7 b,d,f,h).
  7. Added additional links.
  8. Spectra are described in the text. Moving them into the figure caption will make the caption very large.
  9. The analysis procedure is described in more detail.
  10. With optical magnification, only the inclusion of chromite is visible. If there are other inclusions, they are submicron or nanosized. Their study was not included in our plan and requires the use of transmission electron microscopy and other local research methods.
  11. Didn't understand the question. A photomicrograph of a diamond cuboid from the Kumdy-Kol deposit has been added to show the typical growth shape of the cuboid.
  12. Completely agree with the reviewer. We have added relevant links, removed some links to Russian articles.
  13. Completely agree with the reviewer. Corrected the title of the article.
  14. As best I could, I checked the English language and links.
